# Use of Corn Steep Liquor as a Biostimulant in Agriculture

Francisco Garcia-Sanchez [1,*], Jose M. Camara-Zapata [2] and Iván Navarro-Morillo [3]

1   Center for Edaphology and Applied Biology of Segura, Higher Council for Scientific Research, CEBAS-CSIC, 30100 Murcia, Spain
2   Institute for Agro-Food and Agro-Environmental Research and Innovation (CIAGRO-UMH), Miguel Hernández University, 03312 Orihuela, Spain; jm.camara@umh.es
3   Research + Development Department of Atlántica Agrícola, 03400 Villena, Spain; inavarro@atlanticaagricola.com
*   Correspondence: fgs@cebas.csic.es

**Abstract:** Biostimulants are substances or microorganisms that are applied to plants, soil, or seeds, to improve the growth, development, performance, and quality of crops. Their application is mainly directed towards improving the resistance of crops against abiotic and biotic stresses. These compounds are formulated from a great variety of compounds: humic substances, complex organic materials (sewage sludge extracts, composts, and manure), chemical elements (Al, Co, Na, Se, and Si), inorganic salts including phosphite, seaweed extracts (brown, red, and green), amino acids, etc. As of today, it has been observed that corn steep liquor (CSL), which is obtained from the industrial process of corn transformation, may be a good ingredient for the formulation of biostimulant products. CSL contains a large amount of different chemical compounds with biological activity for the plants and soil. The use of CSL industrial waste, previously formulated, could have a direct or indirect effect on the physiological processes and metabolic routes of plants related to the adaptation to abiotic and biotic stresses, as their compounds are part of these metabolic pathways, act as elicitor compounds, and/or have their own biological activity in the plants. There is evidence that the application of CSL could protect plants from specific abiotic and biotic stresses, such as an excess of light or temperature, nutritional limitations, salinity, drought, or pathogens. In addition, it can improve the growth of the plant when these are grown in hydroponic systems, and can improve the health of soils. The present article is focused on describing the most relevant scientific aspects of CSL when used as an ingredient to formulate biostimulants for agriculture. It will discuss its chemical composition, the analytical techniques utilized to elucidate and quantify its compounds, its uses in agriculture, and mode of action in the plants.

**Keywords:** abiotic stress; crop production; primary and secondary metabolites of plants; chemical analysis techniques

## 1. Introduction

Biostimulant products are considered potential and innovative tools in the agricultural sector, as they contain substances that help agronomic crops withstand and adapt to certain abiotic environmental limitations such as drought, salinity, metal heavy toxicity, high temperatures, boron toxicity, plagues, and diseases [1]. The main countries promoting these products within the European Union are France, Italy, and Spain. According to the 2019 report by Grand View Research, Inc., the size of the biostimulant market is expected to reach USD 4.14 billion by 2025, which is an indication of the importance these products will have in the near future [2–4]. These products are formulated with ingredients of complex nature, whose composition encompasses a broad range of molecules.

At the scientific level, plant biostimulants have many definitions, but the most agreed-upon as of today is the following: products obtained from different organic, inorganic,

and/or microorganism-based substances that are able to improve the growth and productivity of plants, and alleviate the negative effects of abiotic stresses [5,6]. The mode of action of these products is difficult to elucidate, due to their complex chemical composition, but diverse mechanisms have been described, such as [7–9]: direct effects on the physiology and metabolism of plants and soil, modification of molecular processes that regulate the primary and secondary metabolic routes related with the efficient use of water and nutrients, vegetative stimulation of the plants, and adaptation to abiotic stresses such as drought, salinity, toxicity due to heavy metals, and boron.

The materials utilized for the production of biostimulants are those that contain mineral elements, vitamins, amino acids, poly- and oligosaccharides, traces of natural hormones from plants, etc. However, the current trend is to use complex materials from industrial waste, which have a great quantity of organic compounds of different nature. Among these materials, one of them is the corn steep liquor extract, for which the state of the art will be revised in this publication.

## 2. General Aspects of Corn Steep Liquor

The world corn production is more than one billion tons a year, representing about 35% of the total world production of cereals. In the corn wet milling production process, the main byproduct is concentrated to produce corn steep liquor (CSL). According to the Corn Refiners Association, "corn steep liquor is a liquid viscous mixture that exclusively contains soluble components in water" [10]. Given that the corn grain is a complex mixture of starch, protein, oil, fiber, minerals, and vitamins, the corn wet milling process produces many products that are widely used in diverse industries, such as corn oil, corn germ flour, corn starch, and high-fructose corn syrup, and the aqueous extract obtained as a residue from these processes can be used as an ingredient in the production of biostimulant products.

The industrial wet manipulation process of corn grain, briefly described here, consists of the soaking of corn for 24–48 h in water containing 0.1–0.2% sulfur dioxide, at 48 to 54 °C. The sulfurous acid created induces chemical and physical changes to the grain, effectively separating the starch and insoluble protein through the breaking of disulfide bonds from the endosperm protein matrix. The sulfurous acid also helps in controlling undesirable organisms and allows the dissolved sugars to convert to lactic acid due to the spontaneous fermentation by lactic bacteria, which helps in the maintenance of the pH close to 4. During the soaking process, different components are separated, such as starch, gluten, and fiber, with only the soaking water remaining, containing a great variety of compounds. This liquid is then evaporated to produce corn syrup [10]. The insoluble corn grain is processed even further, to produce many products utilized in animal feed and/or fertilizers.

CSL is mainly composed of diverse amino acids and reducing sugars [11,12]. The analytical results of this byproduct, related to its content of free and total amino acids, total sugars, reducing sugars, total sulfites, and total nitrogen, verify the importance of CSL as a fermentation substrate, although its physical and chemical functions can be affected by the diverse varieties of corn utilized in the process [12]. Given that the matrix is very complex, it is challenging to develop analytical methods for the identification and quantification of each of the active components that compose it. This complexity of this matrix implies two analytical challenges: the first challenge is to find reliable assay methods, and the second is to develop laboratory protocols that allow for the precise analysis of the target substances without chemical interference [13].

CSL has a wide variety of industrial uses. For example, it can be mixed with fiber, dehydrated and dried as livestock feed, or directly used as nutritional media for industrial fermentation [14,15]. Historically, this byproduct has been used as a source of supplementary nutrients in fermentation processes, livestock feed, production of antibiotics, and as an integral part of fertilizers, growth media, and soil regulators [16–18]. CSL is broadly used as an additive in the technological production of diverse substances such as penicillin, glutamic acid, lactic acid, and glucose oxidase [19,20]. It is often added as an inexpensive source of C-, N-, vitamins, or micronutrients, as an alternative to expensive, established

ingredients in traditional growing media for microorganism culture [21–25]. Some researchers have observed the presence of phenolic compounds in CSL, which have an inhibitory activity against oxidative damage when tested in vitro, which suggests that the consumption of these natural antioxidant compounds can help prevent oxidative damage, and that CSL can be a useful source of phenolic antioxidants [26].

Due to the presence of sulfites, most of the industries cannot completely utilize CSL, and consider it as wastewater that must be treated. As it is considered to be a highly concentrated organic acid residue, its treatment is complicated and expensive. Its consideration as wastewater not only wastes resources, but can also cause great pollution of the environment in case of spills. Determining how to develop and utilize CSL has become an important problem that businesses and society must tackle. In the last few years, CSL has been used as an active material in the production of biostimulant products.

## 3. Active Compounds and CSL Analysis Methods

Given the complex matrix of CSL, it is difficult to develop analytical methods for its identification and quantification, although many scientists are starting to make advances. In the 2010s, diverse techniques were described to analyze and quantify CSL ingredients. Through the use of mass spectrometry, carbohydrates, glycolic acid, lactic acid, and fatty acids can be quantified. Atomic emission spectrometry with inductively coupled plasma and atomic absorption spectrometry allow for the quantification of metallic elements. Total acidity, total fatty acids, sulfur, and total nitrogen have been easily and routinely quantified using near-infrared spectroscopy [13]. Through the selection of the wavelength and more suitable variables, models of partial-least-squares models were fitted, which show good predictive power in the analysis of CSL samples. The experimental data show that near-infrared spectroscopy and partial-least-squares models can replace the traditional chemical analysis method. It is also observed that the total free amino acids in the CSL vary slightly in the samples from different producers, but the types and contents of free amino acids are not very different, among which we find Ile, Asn, Met, Lys, Pro, and Asp. Establishing a principal components model of analysis can facilitate the quality control of CSL [27]. Hofer and Herwiga investigated methods for the determination of vitamins in CSL, and developed a method for the simultaneous analysis of nine different B vitamins through high-performance liquid chromatography [28].

Recently, the nonvolatile and volatile compounds in CSL have been profiled [29]. The first was performed with ultra-high-performance liquid chromatography (UHPLC) equipment coupled to a time-of-flight mass spectrometer (LC/Q-TOF) operating in positive and negative ionization mode. The representative nonvolatile compounds found were 2-Ethyl-2-hydroxybutyric acid, DL-3-Phenyllactic acid, Mannitol, Erythritol, GABA, Isoleucine, and Proline. Of these compounds, it is known that Mannitol, Erythritol, GABA, Isoleucine, and Proline have biological activities as growth promoters and protectors against abiotic and biotic stresses (Table 1).

**Table 1.** Nonvolatile compounds identified in CSL with a known biological activity.

| Chemical Molecule | Chemical Structure | References |
|:---:|:---:|:---:|
| Erythritol |  | [30] |
| Mannitol |  | [31] |
| GABA |  | [32,33] |

**Table 1.** *Cont.*

| Chemical Molecule | Chemical Structure | References |
| --- | --- | --- |
| Isoleucine | | [34,35] |
| Proline | | [36,37] |

The representative volatile compounds found in this study [29] belong to the chemical families Dicarboxylic acids, Fatty acids, Alkanes, Alcohols, Aldehydes, Ketones, Cyclohexanones, Esters, Phenols, Indanones, Methoxybenzenes, Monoterpenoid, and Pyrazines. Of these groups, the most abundant were 1-Decanol, and Ethanone. Phenolic compounds as natural antioxidants can activate enzymes of antioxidant protection which is important for prevention of oxidative damage [38]. These components also have multiple functions in plants, so their high richness in CSL makes this product a good material for the formulation of biostimulants. It must be taken into account that volatile components in plants have multiple functions such as attract pollinators and seed dispersers [39], and induce physiological and metabolic process to cope with biotic and abiotic stresses [40–42]. Therefore, the application of volatile compound in agriculture can have beneficial effects to palliate the negative effects of abiotic and biotic stress in crops [43,44]. These authors describe that the mode of action of these volatile compounds on plants include stabilization of the phospholipid bilayer of chloroplast membranes during exposure to high temperatures, protection of the photosynthetic apparatus and improvement of the water use efficiency in water deficit conditions regulation of oxidative stress through the activation of antioxidant enzymes, activation of cold-protection genes, etc.

In summary, CSL is a rich source of free amino acids, peptides, macroelements, organic acids, polyphenols, and vitamins. This product has a certain variability due to its production methods, but the challenge will be to precisely quantify the largest possible number of compounds to obtain all the scientific information possible to formulate biostimulant products.

## 4. Use of CSL as a Biostimulant to Palliate Abiotic Stresses: Different Case Studies

### 4.1. Improvement of Soil Health

At present, the intensification of agriculture is placing great pressure on natural resources, such as soils. Erosion and soil compacting, loss of nutrients, salinization, chemical contamination, and the depletion of the organic layer affect the microbial community of soils and their ability to retain water and nutrients until they are no longer apt for agriculture. Thus, it has been observed that the application of biostimulant products with CSL are able to regenerate the state of the soils, shown as notable changes in the physicochemical conditions of the treated soil and the essential nutrients such as phosphorus, nitrogen, and potassium. CSL promotes the growth of bacteria that fix nitrogen and solubilize phosphorus [45]. At the same time, this byproduct acts as a biocontrol agent for the pathogens *B. subtilis* and *Pseudomonas* sp. Researchers conducted a study on how the soil microorganisms interact when CSL is applied [46]. The authors harvested nine strains known to be species with a high performance in the biofertilizer industry, including four bacillus species, four fungi, and a yeast species. Using these, three dominant strains were selected according to their ability to dissolve phosphorus, their ability to hydrolyze proteins, and cellular growth. These were *Bacillus subtilis* 3301, *Bacillus licheniformis*, and *Aspergillus Niger*. The fermentation process of these microorganisms was favored when CSL was added to the medium, which indicates that corn liquor waste has an added value when applied to the soil. In places contaminated with hydrocarbons, its application helps with

the degradation of hydrocarbons in chronically contaminated soils. Its application is simple and inexpensive, and it has a great prospective in terms of reutilization, sustainability, and environmental compatibility [47]. These authors studied the effect of CSL on the degradation of hydrocarbons and the structure and function of the microbial community in the microcosm of a humid soil in the field. A chronically contaminated soil treated with CSL was compared with a nontreated soil (3S) for 6 weeks. In the soil treated with CSL, the alipathic and aromatic fractions of hydrocarbons were eliminated, including naphthalene, acetonaphthylene, fluorene, phenanthrene, pyrene, benzo(a)anthracene, and indene(123-cd)pyrene, among others. The metagenomics study revealed that the application of CSL allowed for the development of bacteria with the potential to degrade these compounds, such as *Actinobacteria*, *Verticillium*, *Microbacterium*, *Laceyella*, *Methylosinus*, and *Pedobacter*. The results obtained in that study showed that CSL is a potential resource for the bioremediation of soils contaminated with hydrocarbons.

### 4.2. Stimulation of Plant Growth in Hydroponic Crops

The application of CSL products is not only beneficial for improving soil health to improve crop performance. These types of products can be applied to modern crop systems, known as soilless systems, into a substrate and in floating root systems without any type of substrate. Ref. [10] observed that the use of CSL in hydroponics promotes the production of crops with a healthier shoot, and controls and impedes the appearance of root diseases. It seems that its action is directly related with the promotion of microbial systems and their interactions. Examples of this can be seen in the cultivation of lettuce, in which the use of CSL inhibits the appearance and development of root rot [48]. More recent assays conducted in soilless systems were performed by [49]. These researchers applied a biostimulant product formulated with CSL to pepper plants, to stimulate their growth in a soilless system of peat and perlite. Different parameters of growth, gas exchange, and chlorophyll fluorescence were measured, as well as parameters of oxidative stress, hormones, amino acids and mineral nutrients, and the enzymatic activity of nitrogen metabolism. The results indicated that the application of CSL increased plant growth, as the CSL increased the synthesis of hormones related with cell division and elongation, the net assimilation rate of $CO_2$, and the concentrations of the amino acids alanine, proline, tryptophan, arginine, isoleucine, leucine, and cysteine. The researchers suggested that the mode of action of CSL is related with the regulation and synthesis of hormones and the stimulation of carbon and nitrogen metabolism. Another of the CSL applications described is the improvement in the rate of germination of soy seeds [50]. These researchers observed that 36 h after the application of CSL, as a 1% dose, the germination rate of soy seeds increased, reaching values of 69% as compared to a germination rate of 35% of nontreated seeds. Furthermore, 84 h after the application, the germination rate increased to 95%, beyond the rate of 83% of the control seeds. During the development phase, they also observed beneficial effects, as it increased the pH of the soil and the availability of nutrients in the rhizosphere, resulting in increased vegetative growth and an earlier flowering of the soy plants. Additional studies have shown that the expression levels of some genes related to nutrient absorption and transport in the soy plant roots treated with 1% CSL were significantly higher than the control plants [47]. All of these results show that CSL can be used both as a potential stimulant and fertilizer to increase the growth rate of soy plants.

### 4.3. Reduction in the Use of Conventional Nitrogen Fertilizers

Excessive nitrogen fertilization is causing severe environmental problems related with the eutrophication of natural resources such as oceans, aquifers, and coastal lakes such as the Mar Menor (Spain), among others. Therefore, it is necessary to design agronomic strategies that allow us to grow plants below their nutritional requirements, especially regarding nitrogen. The authors of [29] conducted an experiment to discover if the foliar and root application of CSL extract to pepper plants (*Capsicum annuum* L.), grown with different concentrations of nitrogen (100% to 25% N), could palliate the effects from nitrogen

limitations. At the end of the experiment, the plants were assessed by measuring different parameters such as vegetative growth, nitrate, ammonium, total nitrogen concentration in leaves, concentration of amino acids, soluble proteins, and enzymatic activities of glutamine synthetase and nitrate reductase. The results showed that plants grown with limited nitrogen were very sensitive to this deficiency, and that the exogenous application of CSL was not able to palliate this deficiency. However, it was observed that plants grown under limited nitrogen conditions, to which CSL was applied, improved their nitrogen efficiency parameters (NUtE and NUE, nitrogen utilization efficiency and nitrogen use efficiency, respectively), and this was due to the increase in certain enzymes related with the assimilation of nitrogen in the plant (enzymatic activities of nitrate reductase and glutamine synthetase), and to the increase in the concentration of amino acids and proteins.

### 4.4. Protection against Sunburn

Sunburn is an important problem that affects the performance of many crops, mainly in arid and semiarid regions. Excessive solar radiation and high temperatures can reduce growth and cause foliar chlorosis, oxidative stress, and impediments in photosynthesis. The company Atlántica Agrícola has developed a product named Archer® Eclipse, formulated with CSL and calcium and zinc mineral salts, whose foliar application to cucumber plants (*Cucumis sativus* L.) prevents damage from leaf burns [51]. In a growth chamber, the researchers subjected the plants to sun and heat stress with a high-power sodium lamp placed 90 cm above the plants. The effect of this product was analyzed by evaluating sunburn symptoms on the leaves, vegetative growth parameters, leaf temperature, photosynthesis, and oxidative stress. This study showed that the use of the CSL-based biostimulant produced less sunburn, increased leaf biomass by 43%, decreased the leaf temperature by 3 °C, increased photosynthesis, water use efficiency, and chlorophylls, and decreased oxidative stress biomarkers. The researchers established that the mode of action of this product was due to the protecting effect of the biostimulant, as it was able to reflect some of the light that reached the leaf, and on the other, avoid the harmful effects of the incident light, due to the antioxidant properties of the biostimulant. Therefore, the study confirmed the efficiency of Archer® Eclipse on the protection of sensitive plants against conditions that induce sunburns.

### 4.5. Increase Tolerance to Salinity

Most of the salinity problems in agriculture are caused by the use of irrigation water with a high concentration of salts, mainly chlorides, sodium, and sulfates, the excessive use of fertilizers, or the loss of the leaching capacity of some soils. Its incidence provokes osmotic and specific effects, as well as an imbalance in the absorption of nutrients, which makes the growth of most crops to be difficult. Ref. [52] showed that the application of a biostimulant with CSL increased the tolerance of pepper plants (*Capsium annuum* L.) to salinity when grown with water containing 100 mM NaCl. The authors reported that the beneficial effect of CSL in the reduction in the phytotoxicity of salt stress was due to an improvement in the photosynthetic efficiency of the leaves and the reduction in reactive oxygen species produced by Cl and/or Na toxicity. These results were supported by the concentration of MDA and $H_2O_2$ in pepper plant leaves treated with salinity, being 3.5 times lower when CSL was applied to the root. The results showed that the application of CSL increased the shoot biomass and leaf area in salt conditions through physiological mechanisms, and this beneficial effect was greater when the biostimulant was applied through the root.

Other researchers, ref. [53] applied a combination of myo-inositol and CSL to cabbage seedlings (*Brassica rapa* subsp. pekinensis) grown under saline conditions of 150 mM NaCl. The authors observed that this concentration was very toxic to the plants, but the combined application of myo-inositol and CSL was able to counteract the effects of salinity, resulting in an increase in vegetative growth. This was due to the application of the biostimulant improving photosynthesis, and also maintaining the balance between the osmotic and

ionic components of the cells. The more pronounced positive effects were observed in the treatment with 0.1 mL/L CSL + 288 mg/L myo-inositol.

## 5. Conclusions and Future Perspectives

Corn liquor (CSL) has been highlighted as a versatile byproduct of wet milling of corn, with a composition rich in amino acids, sugars, minerals, and other beneficial compounds. One of its utilities has been to the manufacturing of biostimulant products. CSL has been shown to be efficient in the improvement of soil health, the stimulation of plant growth, the reduction in the use of conventional nitrogen fertilizers, the protection against sunburn, and the increase in salinity tolerance, and this is due to the number of compounds with biological capacity that it contains that act together thanks to their synergies.

The future perspectives of the use of CSL as a biostimulant are promising. The ability of CSL to regenerate soil health, promote plant growth, and improve the use of nutrients could significantly contribute towards agricultural sustainability. Also, its application in conditions of abiotic stress, such as salinity and sunburn, demonstrates its considerable potential for addressing environmental and climactic challenges in agriculture. It is hoped that studies continue to explore new biostimulant formulations based on CSL, as well as more efficient methods for its application. The optimization of doses, frequency, and application methods could maximize its agronomic benefits and minimize any negative environmental impact. Also, additional studies about the molecular mechanisms responsible for the effects of CSL on plants could provide a deeper understanding and promote the continuous improvement of biostimulant products. In the context of agricultural sustainability and the efficient management of resources, CSL could play a key role when offering more sustainable alternatives to conventional supplies. However, it is important to address the challenges associated with the treatment of CSL as a high concentration acidic organic residue to avoid environmental contamination and to maximize its value.

**Author Contributions:** Conceptualization, F.G.-S.; methodology, F.G.-S.; validation, J.M.C.-Z. and I.N.-M.; investigation, F.G.-S.; resources, F.G.-S. and I.N.-M.; writing—original draft preparation, F.G.-S.; writing—review and editing, F.G.-S.; visualization, J.M.C.-Z.; supervision, F.G.-S.; project administration, I.N.-M.; funding acquisition, F.G.-S. and I.N.-M. All authors have read and agreed to the published version of the manuscript.

**Funding:** This article has been subsidized by the company Atlantica Agricola and by the CEBAS-CSIC research center of the Ministry of Sciences, Innovation and Universities of the Government of Spain.

**Data Availability Statement:** Not applicable.

**Conflicts of Interest:** The authors declare no conflicts of interest.

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
