# Peer review of "Use of Corn Steep Liquor as a Biostimulant in Agriculture"

_horticulturae, doi:10.3390/horticulturae10040315_

Round 1

Reviewer 1 Report

Comments and Suggestions for Authors

Dear Authors

The manuscript titled 'Use of Corn Steep Liquor as a Biostimulant in Agriculture' cannot be further processed because it has many serious flaws. My comments are as follows:

1. The article's title does not correspond to the manuscript's content. Generally speaking, the topic is current and essential in agriculture, but the selection of content is inappropriate.

2. The abstract is too extended and has minimal relation to the title of the manuscript.

3. In the Introduction, the division of biostimulants is outdated. In the In the introduction chapter, the division of biostimulants is outdated. The authors of the manuscript did not take into account current legal acts regulating the nomenclature of biostimulants (e.g. ...) (e.g. Regulation (EU) 2019/1009 of the European Parliament and of the Council of 5 June 2019 laying down rules on the making available on the market of EU fertilising products; Regulation (EC) No 1107/2009 of the European Parliament and of the Council of 21 October 2009 concerning the placing of plant protection products on the market)

4. Too much attention was paid to other biostimulants (in the chapters Introduction and Type of Biostimulants), which are not the subject of this review and are widely described in other review articles.

5. The most important chapter, " Use of CSL as a biostimulant to palliate abiotic stresses. Different case studies", which should be the key one in the manuscript, is the summary of only nine articles. The results described in this section can be easily found in the original versions of these publications. In the review article, the authors should refer critically to already published results and present them with their assessment.

6. The most important chapter, "Use...", which should be a crucial part of the article, is the summaries of only nine articles, including those by the same authors as the reviewed manuscript. Moreover, the results described in this chapter can be easily found in the original versions of these articles. In the review, the authors should refer critically to already published results and present them with their assessment. Therefore, using many scientific reports to verify scientific facts reliably is very important. However, if research in a given area is sparse, it may be too early to review the specific topic.

Author Response

The manuscript titled 'Use of Corn Steep Liquor as a Biostimulant in Agriculture' cannot be further processed because it has many serious flaws. My comments are as follows:

  1. The article's title does not correspond to the manuscript's content. Generally speaking, the topic is current and essential in agriculture, but the selection of content is inappropriate.

Authors: The ms has bee shorted and focused in CSL.

  1. The abstract is too extended and has minimal relation to the title of the manuscript.
  2. In the Introduction, the division of biostimulants is outdated. In the In the introduction chapter, the division of biostimulants is outdated. The authors of the manuscript did not take into account current legal acts regulating the nomenclature of biostimulants (e.g. ...) (e.g. Regulation (EU) 2019/1009 of the European Parliament and of the Council of 5 June 2019 laying down rules on the making available on the market of EU fertilising products; Regulation (EC) No 1107/2009 of the European Parliament and of the Council of 21 October 2009 concerning the placing of plant protection products on the market).

Authors. We have opted for a scientific nomenclature as indicated by another expert reviewer.

  1. Too much attention was paid to other biostimulants (in the chapters Introduction and Type of Biostimulants), which are not the subject of this review and are widely described in other review articles.

Authors: This has been corrected. Manuscript has been focused in CSL.

  1. The most important chapter, " Use of CSL as a biostimulant to palliate abiotic stresses. Different case studies", which should be the key one in the manuscript, is the summary of only nine articles. The results described in this section can be easily found in the original versions of these publications. In the review article, the authors should refer critically to already published results and present them with their assessment.

Authors: We have redone the ms from a review to an minireview.

  1. The most important chapter, "Use...", which should be a crucial part of the article, is the summaries of only nine articles, including those by the same authors as the reviewed manuscript. Moreover, the results described in this chapter can be easily found in the original versions of these articles. In the review, the authors should refer critically to already published results and present them with their assessment. Therefore, using many scientific reports to verify scientific facts reliably is very important. However, if research in a given area is sparse, it may be too early to review the specific topic.

Authors: We have redone the ms from a review to an minireview.

Reviewer 2 Report

Comments and Suggestions for Authors

Dear Authors,

This review article covers topic regarding bio stimulant used in agriculture with the emphasize on corn steep liquor. Manuscript is well organized and good covers main topic. There is few suggestions which have to be corrected.

In the section introduction you are mentioning biotic and abiotic environmental limitation. Highlight in manuscript on which limitations you meant.

Give a detailed chemical characterization of corn steep liquor. Highlight in in manuscript.

Which specific phenolic compounds are discovered in corn steep liquor. Insert in in manuscript.

Highlight in manuscript that phenolic compounds as a natural antioxidants can activate enzymes of antioxidant protection which is important for prevention of oxidative damage. Kindly consider to cite Agronomy, 11(7), (2021) 1414.

Conclusion is too long. Make it more direct and do not repeat which you already mentioned before.

Wish you all the best in the future work,

Author Response

This review article covers topic regarding bio stimulant used in agriculture with the emphasize on corn steep liquor. Manuscript is well organized and good covers main topic. There is few suggestions which have to be corrected.

In the section introduction you are mentioning biotic and abiotic environmental limitation. Highlight in manuscript on which limitations you meant.

Authors: This limitation has been introduced in the “Introduction”.

Give a detailed chemical characterization of corn steep liquor. Highlight in in manuscript.

Which specific phenolic compounds are discovered in corn steep liquor. Insert in in manuscript.

Authors: chemical characterization has been discussed throughout the manuscript

Highlight in manuscript that phenolic compounds as a natural antioxidants can activate enzymes of antioxidant protection which is important for prevention of oxidative damage. Kindly consider to cite Agronomy, 11(7), (2021) 1414.

Authors: These references has been included.

Conclusion is too long. Make it more direct and do not repeat which you already mentioned before.

Authors: Conclusion has been shorted.

Reviewer 3 Report

Comments and Suggestions for Authors

The review paper entitled “Use of Corn Steep Liquor as a Biostimulant in Agriculture” fits with the general scope of the Journal Horticulturae MDPI. In this review, the authors collected information about the possibility to employing corn steep liquor as biostimulant for agricultural crops. The manuscript is well written, however, I believe that there are some flaws that need to be addressed before publication. I suggest the authors to revise the entire structure of the review. The paper could be interesting, however, there is no point in mentioning all the other biostimulants if the subject is the corn steep liquor biostimulant. A small mention of all biostimulants is a must (types, main effects etc.), but the focus must be on maize liquor. If there is little on the biostimulant in question (as it would seem since it is a new product), a mini review would be more appropriate since, in my opinion, it is a bit short (15 pages reference included). Morevover, since the biostimulant is relatively new, it would be interesting (if the authors find information) to include the economic viability of producing such a biostimulant (how much it costs vs. how much economic benefit it provides). Consequently, I suggest the authors to search for more information on the biostimulant, remove too much information on other biostimulants and change the type of manuscript from review to mini-review. For all above, I consider the paper suitable for publication only after MAJOR revisions.

Author Response

The review paper entitled “Use of Corn Steep Liquor as a Biostimulant in Agriculture” fits with the general scope of the Journal Horticulturae MDPI. In this review, the authors collected information about the possibility to employing corn steep liquor as biostimulant for agricultural crops. The manuscript is well written, however, I believe that there are some flaws that need to be addressed before publication.

 I suggest the authors to revise the entire structure of the review. The paper could be interesting, however, there is no point in mentioning all the other biostimulants if the subject is the corn steep liquor biostimulant. A small mention of all biostimulants is a must (types, main effects etc.), but the focus must be on maize liquor. If there is little on the biostimulant in question (as it would seem since it is a new product), a mini review would be more appropriate since, in my opinion, it is a bit short (15 pages reference included). Morevover, since the biostimulant is relatively new, it would be interesting (if the authors find information) to include the economic viability of producing such a biostimulant (how much it costs vs. how much economic benefit it provides). Consequently, I suggest the authors to search for more information on the biostimulant, remove too much information on other biostimulants and change the type of manuscript from review to mini-review. For all above, I consider the paper suitable for publication only after MAJOR revisions.

Authors: We have taken account all these comments, and have rewriiten the ms with this comments.

Reviewer 4 Report

Comments and Suggestions for Authors

The authors reviewed the use of CSL as a biostimulant. The text suffers from the lack of focus, weak presentation, and sometimes superficial treatment. I recommend to prune irrelevant text, deepen the discussion of the relevance of facts collected from literature, and focus on the topic. This especially holds for the Abstract and Conclusions. Examples:

INACCURATE/MISLEADING STATEMENTS

Abstract: "These compounds, as opposed to conventional mineral fertilizers, are formulated from a great variety of compounds, among which we find –Substances and a mix of materials, -plants, plant parts or plant extracts, - Compost, -Fresh crop digestate, -Digestate other than fresh crop digestate, Food industry by-products, microorganisms, Nutrient polymers, Polymers other than nutrient polymers, etc." The authors emphasize that biostimulants are mainly used to counter biotic and abiotic stress; therefore, the counterpart for a comparison are pesticides and agents of biological control rather than "conventional mineral fertilizers". (The fact that a particular legal text lists biostimulants among "fertilizing products" does not justify this grouping in a scientific narrative; the function of biostimulants is fundamentally different from mineral fertilizers.) Furthermore, the sentence would not pass even the laxest standard of language quality.  

IRRELEVANT STATEMENTS

Abstract: "CSL contains a large amount of different chemical compounds belonging to both primary and secondary plant metabolism, as shown by the use of different analytical techniques such as mass spectroscopy, inductively coupled plasma mass spectrometry, atomic absorption spectroscopy, Ultra- High Performance Liquid Chromatography (UHPLC) coupled to a Time-of-Flight Mass Spectrometer, etc." The list contains a random selection of methods at different hierarchic levels and as such is irrelevant. For instance, MS is mentioned as a method of its own, and additionally as TOF part of hyphenated LC. The reader wonders what does it mean and why other relevant methods are missing. Furthermore, capitalization is wrong.

Introduction: The authors delve on diverse products that have nothing to do with CSL like limestone, manure, microorganisms, compost, or precipitated phosphate salts. Lines 48 – 62 are irrelevant to the topic of the manuscript.

Similarly, lines 79 – 173 (2 full pages), rather than providing a contextual background to CSL, deal with biostimulants other than CSL.

Many passages, short and long, describe facts that are irrelevant to the topic and often trivial, such as this: "The volatile compounds emitted by plants include substances from diverse metabolic origins such as alkaloids, phenols, nitrogen-containing compounds, terpenoids, and volatiles derived from fatty acids, known as green leaf volatiles." This superficial excursion to the chemistry of plant VOCs only dilutes the narrative and distracts from the topic.

Another example of an inaccurate statement: "CSL is mainly composed of diverse amino acids and reducing sugars [46,47]." Elsewhere in the text, the authors explain that CSL "is rich in B vitamins and minerals", and at another point they write about "the presence of phenolic compounds in CSL". Statements about the composition of CSL are scattered through the manuscript rather than centralized in a section focusing on the subject.

SOME REFERENCES ARE INADEQUATE

"CSL is mainly composed of diverse amino acids and reducing sugars [46,47]." Paper. 46 describes the effect of Ca ions on starch-water system. It does not even mention corn steep liquor!

"CSL as a fermentation substrate, although its physical and chemical functions can be affected by the diverse varieties of corn utilized in the process [48]." Ref. 48 is on the use of CSL as feed in aquaculture; it does not provide any data on the effect of corn variety on the properties of CSL.

"...this by-product acts as a biocontrol agent for the pathogens B. subtilis and Pseudomonas sp. [81] this by-product acts as a biocontrol agent for the pathogens B. subtilis and Pseudomonas sp. [81]" Support for the use in biocontrol is missing, paper 81 deals with the use of CSL as substrate in fermentation.

ABSTRACT AND CONCLUSIONS HAVE TO BE FOCUSED

Repeating the uninformative sentence "The complexity of the CSL matrix has resulted in analytical challenges, but the advances in the techniques of mass spectrometry, atomic emission spectrometry, near infrared spectroscopy, and other methods, has allowed for the identification and quantification of its active components." is out of place in the Abstract and Conclusion (it also appears in the main text). In the Conclusions, the statement appears after the composition of CSL was described as being "rich in amino acids, sugars, minerals, and other beneficial compounds." The reader is curious to know which compounds. He/she will be disappointed finding out that what follows is a trivial list of general analytical methods instead of specific information about CSL.

PRESENTATION OF DATA

The unreadable text on L264 – 280 should be replaced by a different kind of presentation. The related Figure 1 with VOCs concealed by numbers, in its current form, is useless: nobody wants to search for the numbers L264 – 280, and even if someone does, they will miss the whole picture. The authors should replace the text and Fig. 1 but an informative presentation of the finding and explain what these data tell us.

Comments on the Quality of English Language

The quality of English language is adequate except for capitalization, I pointed out some of the problems above. Furthermore, one passages is unreadable (see comment above). The cause is not English quality but the choice of the way how the results are presented. There are many errors in capitalization in this particular passage, however. Note that the names of chemicals and analytical methods, except for abbreviations, are not capitalized.

Author Response

The authors reviewed the use of CSL as a biostimulant. The text suffers from the lack of focus, weak presentation, and sometimes superficial treatment. I recommend to prune irrelevant text, deepen the discussion of the relevance of facts collected from literature, and focus on the topic. This especially holds for the Abstract and Conclusions.

INACCURATE/MISLEADING STATEMENTS

Abstract: "These compounds, as opposed to conventional mineral fertilizers, are formulated from a great variety of compounds, among which we find –Substances and a mix of materials, -plants, plant parts or plant extracts, - Compost, -Fresh crop digestate, -Digestate other than fresh crop digestate, Food industry by-products, microorganisms, Nutrient polymers, Polymers other than nutrient polymers, etc."

The authors emphasize that biostimulants are mainly used to counter biotic and abiotic stress; therefore, the counterpart for a comparison are pesticides and agents of biological control rather than "conventional mineral fertilizers". (The fact that a particular legal text lists biostimulants among "fertilizing products" does not justify this grouping in a scientific narrative; the function of biostimulants is fundamentally different from mineral fertilizers.) Furthermore, the sentence would not pass even the laxest standard of language quality.

Authors: I agree with this comment. Abstracts has been rephrased.

IRRELEVANT STATEMENTS

 Abstract: "CSL contains a large amount of different chemical compounds belonging to both primary and secondary plant metabolism, as shown by the use of different analytical techniques such as mass spectroscopy, inductively coupled plasma mass spectrometry, atomic absorption spectroscopy, Ultra- High Performance Liquid Chromatography (UHPLC) coupled to a Time-of-Flight Mass Spectrometer, etc." The list contains a random selection of methods at different hierarchic levels and as such is irrelevant. For instance, MS is mentioned as a method of its own, and additionally as TOF part of hyphenated LC. The reader wonders what does it mean and why other relevant methods are missing. Furthermore, capitalization is wrong.

Authors: This sentence has been deleted in the ms.

4) Introduction: The authors delve on diverse products that have nothing to do with CSL like limestone, manure, microorganisms, compost, or precipitated phosphate salts. Lines 48 – 62 are irrelevant to the topic of the manuscript.

Authors: This sentence has been deleted in the ms.

5) Similarly, lines 79 – 173 (2 full pages), rather than providing a contextual background to CSL, deal with biostimulants other than CSL.

Authors: This sentence has been deleted in the ms.

6) Many passages, short and long, describe facts that are irrelevant to the topic and often trivial, such as this: "The volatile compounds emitted by plants include substances from diverse metabolic origins such as alkaloids, phenols, nitrogen-containing compounds, terpenoids, and volatiles derived from fatty acids, known as green leaf volatiles." This superficial excursion to the chemistry of plant VOCs only dilutes the narrative and distracts from the topic.

Authors: Sentences about VOCs has been shorted.

7) SOME REFERENCES ARE INADEQUATE

"CSL is mainly composed of diverse amino acids and reducing sugars [46,47]." Paper. 46 describes the effect of Ca ions on starch-water system. It does not even mention corn steep liquor!

"CSL as a fermentation substrate, although its physical and chemical functions can be affected by the diverse varieties of corn utilized in the process [48]." Ref. 48 is on the use of CSL as feed in aquaculture; it does not provide any data on the effect of corn variety on the properties of CSL.

"...this by-product acts as a biocontrol agent for the pathogens B. subtilis and Pseudomonas sp. [81] this by-product acts as a biocontrol agent for the pathogens B. subtilis and Pseudomonas sp. [81]" Support for the use in biocontrol is missing, paper 81 deals with the use of CSL as substrate in fermentation.

Authors: All references has been reviewed again.

These references has been corrected [46,47,48]:

Xue Xiao, Yuanyuan Hou, Yang Liu, Yanjie Liu, Hongzhi Zhao, Linyi Dong, Jun Du, Yiming Wang, Gang Bai, Guoan Luo, Classification and analysis of corn steep liquor by UPLC/Q-TOF MS and HPLC, Talanta,Volume 107, 2013, 344-348, https://doi.org/10.1016/j.talanta.2013.01.044.

Zhou, K., Yu, J., Ma, Y. et al. Corn Steep Liquor: Green Biological Resources for Bioindustry. Appl Biochem Biotechnol 194, 3280–3295 (2022). https://doi.org/10.1007/s12010-022-03904-w

[81]. These references has been placed in a correct place.

8) ABSTRACT AND CONCLUSIONS HAVE TO BE FOCUSED

Repeating the uninformative sentence "The complexity of the CSL matrix has resulted in analytical challenges, but the advances in the techniques of mass spectrometry, atomic emission spectrometry, near infrared spectroscopy, and other methods, has allowed for the identification and quantification of its active components." is out of place in the Abstract and Conclusion (it also appears in the main text). In the Conclusions, the statement appears after the composition of CSL was described as being "rich in amino acids, sugars, minerals, and other beneficial compounds." The reader is curious to know which compounds. He/she will be disappointed finding out that what follows is a trivial list of general analytical methods instead of specific information about CSL.

Authors: Abstract and conclusion has been rewritten.

9) PRESENTATION OF DATA

The unreadable text on L264 – 280 should be replaced by a different kind of presentation. The related Figure 1 with VOCs concealed by numbers, in its current form, is useless: nobody wants to search for the numbers L264 – 280, and even if someone does, they will miss the whole picture. The authors should replace the text and Fig. 1 but an informative presentation of the finding and explain what these data tell us.

Authors: Figure 1 has been eliminated.

Round 2

Reviewer 1 Report

Comments and Suggestions for Authors

Dear Authors

According to the information available on the Horticulturae website, there is no such form of publication as Minireview. In turn, there is a Review article type, which must be prepared in accordance with PRISMA guidelines. In my opinion, the article in its current form does not meet these requirements.

Reviewer 2 Report

Comments and Suggestions for Authors

Dear Authors, 

Thank you very much for revised version of your manuscript and all answers, it is fine for me. 

Wish you all the best in future, 

Reviewer 3 Report

Comments and Suggestions for Authors

The authors address all point of criticism raised by me. Consequently, I consider the manuscript suitable for publication.

Reviewer 4 Report

Comments and Suggestions for Authors

The manuscript has been substantially improved, yet I still recommend the authors to consider the following:

L15: formulated from a variety of compounds:... chemical elements (Al, Co, Na, Se, and Si)
Comment: Elementary Na does not occur in nature, as it would immediately react with water, producing sodium hydroxide. Co, Al, Se, and Si also do not occur in elementary form in nature.

L16: inorganic salts including phosphite,
Comments: Check whether you have not meant phosphate.

L136-138: check capitalization

L142-144: check capitalization

Comments on the Quality of English Language

English is fine, except for minor issue such as capitalization (see my comments for examples). The final proofreading at MDPI office suffices to correct these errors.